# Styletubation in Bariatric Surgery: A Case Report

**DOI:** 10.3390/healthcare11162256

**Published:** 2023-08-10

**Authors:** Bor-Gang Wu, Hsiang-Ning Luk, Jason Zhensheng Qu, Alan Shikani

**Affiliations:** 1Department of Surgery, Hualien Tzu-Chi Hospital, Buddhist Tzu Chi Medical Foundation and School of Medicine, Tzu-Chi University, Hualien 970, Taiwan; brogen.bgw@gmail.com; 2Department of Anesthesia, Hualien Tzu-Chi Hospital, Hualien 970, Taiwan; 3Bio-Math Laboratory, Department of Financial Engineering, Providence University, Taichung 433719, Taiwan; 4Department of Anesthesia, Critical Care and Pain Medicine, Massachusetts General Hospital, Harvard Medical School, Boston, MA 02115, USA; jqu@mgh.harvard.edu; 5Division of Otolaryngology—Head and Neck Surgery, LifeBridge Sinai Hospital, Baltimore, MD 21040, USA; ashikani@gmail.com; 6Division of Otolaryngology—Head and Neck Surgery, MedStar Union Memorial Hospital, Baltimore, MD 21218, USA

**Keywords:** styletubation, video-assisted intubating stylet, obesity, super-super obesity, bariatric surgery, laparoscopic sleeve gastrectomy, tracheal intubation, laryngoscopy, videolaryngoscope, anesthesia, difficult airway

## Abstract

Direct laryngoscopes and videolaryngoscopes are the dominant endotracheal intubation tools. The styletubation technique (using a video-assisted intubating stylet) has shown its advantages in terms of short intubation time, high success rate, less required stimulation, and operator satisfaction. The learning curve can be steep but is easily overcome if technical pitfalls are avoided. Conditions that make styletubation challenging include secretions/blood, short/stiff neck, restricted mouth opening and cervical spine mobility, anatomical abnormalities over head and neck regions, obesity, etc. In this clinical report, we present the effectiveness and efficiency of the routine use of the styletubation for tracheal intubation in a super-super-obese patient (BMI 103 kg/m^2^) undergoing bariatric surgery with laparoscopic sleeve gastrectomy.

## 1. Introduction

Obesity, an increasing medical problem worldwide, is measured through body mass index (BMI) > 30 kg/m^2^ and further categorized into grade 1 (BMI 30 to <35 kg/m^2^), grade 2 (BMI 35 to <40 kg/m^2^), and grade 3 (BMI ≥ 40 kg/m ^2^). The prevalence of super obesity (SO, BMI > 50.0 kg/m^2^) and super-super obesity (SSO, BMI > 60 kg/m^2^) has also increased during recent years. Bariatric surgery is the most effective therapy to treat this and is sometimes the last resort in efforts to lose weight. Expectedly, anesthesia and peri-operative management for bariatric and non-bariatric surgeries in all categories of the obese patient populations have become focus issues up for discussion [1,2,3,4]. Among all the relevant peri-operative management, airway management in such obese populations has been extensively discussed in the literature. Airway management, particularly intubation, remains the most challenging part of intraoperative care [5,6,7,8,9,10,11,12,13].

The severity of obesity is related to the incidence of difficult airway [14], although there some controversies surrounding this [15]. The disagreement may arise from the different criteria for defining difficult intubation. Examples of the criteria for predicting difficult intubation are body mass index (BMI), neck circumference, degree of neck mobility, width of mouth opening, etc. [16]. Different definitions for difficult airway were used, such as difficult laryngoscopy (e.g., impossible to visualize any portion of the vocal cords after multiple attempts at laryngoscopy) or difficult tracheal intubation (e.g., requires multiple attempts to intubate) [17]. Others might use the intubation difficult scaly (IDS) or its modified version, combined with additional parameters, to predict/identify the occurrence of difficult airway.

Since videolaryngoscopes (VL) were invented over two decades ago, the technology has been repeatedly compared with conventional direct laryngoscopes (DL). VL has been shown to possess certain advantages over the DL, including better glottic visualization, less need for the alignment of the airway axes, less force and cervical spine manipulation, shorter intubation time, and perhaps a higher first-pass success rate of intubation [18,19]. A plethora of literature show the advantages (e.g., easier, faster, less complications) of using VL in obese populations [20,21,22,23,24,25,26,27,28,29,30]. However, the superiority of VL over DL, regarding all the outcome parameters (e.g., speed, safety, visualization, easiness), has not always been consistently confirmed in the obese patient population [31,32,33,34].

Various alternative intubation tools have been reported other than DL and VL [35]. Those tracheal intubating modalities include rigid Bullard^TM^ laryngoscope, flexible fiberoptic bronchoscope (FOB), optical stylet [36,37,38,39], or combined FOB with VL [40]. The video-assisted intubating stylet technique has recently been termed “styletubation”, in contrast to the concept of the conventional laryngoscopy [41]. With extensive experience of using styletubation recently in Taiwan, clinicians found this technique useful and easy to learn. In this case report, we present our experience of applying styletubation in a super-super-obese patient undergoing laparoscopic sleeve gastrectomy.

## 2. Case Presentation

A 33-year-old man (height: 158 cm, weight: 258 kg, and body mass index (BMI): 103 kg/m^2^) was referred to our Center for Obesity & Metabolic Health, Hualien Tzu-Chi Medical Center. After entering a six-month program (medical weight loss, education, and counseling) that included treatment with orlistat and liraglutide, his body weight had been successfully reduced down to 227 kg (BMI, 90.9 kg/m^2^). Bioelectric impedance measurements showed the breakdown of the patient’s body composition: percentage body fat 54.1%, waist 200 cm, and hip 202 cm. Laparoscopic sleeve gastrectomy was scheduled. Medical history included hypertension (blood pressure 205/100 mmHg, heart rate 75 beats/min) and cellulitis over lower extremities. Losartan and amlodipine were prescribed in order to control high blood pressure. Pre-operative physical check-ups were conducted, including: a transthoracic echocardiogram, which revealed trivial tricuspid regurgitation, normal left ventricular motion (ejection fraction, 69%), no abnormal regional wall motion, and an estimated pulmonary artery pressure = 24 mmHg; a pulmonary function test, which revealed mild obstructive lung disease; abdominal echography, which revealed marked fatty liver; esophagogastroduodenoscopy, which revealed esophagitis and gastroesophageal reflux disease; and polysomnography, which detected moderate obstructive sleep apnea (apnea/hyponea index: 22.5/h). The blood panel showed HbA1c 5.9% and hemoglobin 11.5 g/dL.

Pre-operative airway evaluation showed a neck circumference of 54 cm; interincisor distance of 4.5 cm; sternomental distance of 17 cm; thyromental distance of 8 cm; and Mallampati class IV (Figure 1). The observed apnea, high blood pressure, body mass index, age, neck circumference, and gender (STOP-BANG) created a score of 7. Body weights by definition were as follows: actual BW 227 kg, ideal BW 55 kg, adjusted BW 124 kg, and lean body weight 103 kg. Before the induction of anesthesia, the patient was placed on the operating table in a reverse Trendelenburg with ramp position (Figure 2). His head and torso were elevated such that his external auditory meatus and the sternal notch were approximately horizontally aligned. He received pure oxygen at a rate of 30 L/min with a high-flow nasal cannula for 15 min. The patient was monitored, as recommended by the American Society of Anesthesiologists’ (ASA) standards, using pulse oximetry (SpO_2_), capnography (ETCO_2_), electrocardiogram (ECG), non-invasive (NIBP) and invasive arterial blood pressure examinations (A-line), neuromuscular blockade monitoring (train-of-four, TOF), bispectral index (BIS) and density spectral array (DSA) monitoring, a minimally invasive FloTrac system, cerebral oximetry, and nociception monitoring (Surgical Plethysmographic Index, SPI).

Pre-induction vital signs were blood pressure, 140/78 mmHg; heart rate, 53 per minute; respiratory rate, 16 per minute; and SpO_2_, 100%. The induction of anesthesia was started using glycopyrrolate 0.2 mg, lidocaine 50 mg, and midazolam 5 mg, followed by the administration of propofol 180 mg and rocuronium 100 mg. An airway assistant performed the jaw-thrust maneuver and opened the patient’s mouth. A nasal–pharyngeal airway with a suction tube was inserted into the patient’s oropharyngeal space to clear the secretions. It also served as a guide for subsequent intubation. Tracheal intubation was performed via the styletubation technique (video-assisted intubating stylet). A standard endotracheal tube with 7.5 mm of internal diameter was used for intubation (Figure 3). The whole process was smooth (first-pass success) and swift (12 s) (Figure 4). The airway was then assessed by the capnography and chest auscultation. The ventilator strategy and parameters were set as follows: pressure control ventilation volume guaranteed (PCV-VG) with tidal volume: 650 mL; positive end-expiratory pressure (PEEP): 7 cm H_2_O; respiratory rate: 14–16 breaths/min; fresh gas flow rate: 2 L/min; fraction of inspired oxygen: 60%; maintenance partial pressure of end-tidal carbon dioxide: 33–45 mmHg; and peak pressure: 28–33 mmHg.

Anesthesia was maintained using propofol (under target-controlled infusion, TCI Marsh model, Ce 2.0 μg/mL) and desflurane 4–6% (MAC 0.6–0.8, BIS 40–50). Fentanyl and rocuronium were supplemented based on the SPI and TOF values. In brief, the patient was then placed in a supine position and the 5-trocar method was used. A vertical gastrectomy was performed via resection of the greater curvature from the distal antrum (5 cm proximal to the pylorus) to the angle of His, using a 36-French bougie as a calibration tube. The resected part of the stomach was then extracted from the periumbilical trocar site. The staple line was reinforced using a running absorbable seromuscular suture. The anesthetic and surgical procedure lasted for 5 h. The SPI value from the intra-operative analgesic monitoring was monitored and remained around 50. During the emergence phase, desflurane was stopped while dexmedetomidine was maintained at 0.2 μg/kg/h. The patient regained spontaneous breath 2 min after the injection of sugammadex (400 mg). Then, dexmedetomidine infusion was stopped and he was extubated smoothly. The patient stayed in the intensive care unit overnight. Post-operative analgesia was achieved using parecoxib (40 mg, intravenous) and transversus abdominis plane block (TAP block, 40 mL of 0.5% ropivacaine). The prophylactic use of dexamethasone was performed. In the ICU, the patient reported no significant pain and did not request any rescue analgesia. Since the patient had OSAS, continuous positive airway pressure (CPAP) was applied overnight. This did not reveal any dyspnea, respiratory depression, or hypoxia. No post-operative pulmonary complications, e.g., respiratory depression, pneumonia, hypoxia, upper airway obstruction, were reported. After 24 h in the ICU, the patient was discharged to the ward and started receiving the rehabilitation program. On the 8th postoperative day, he was discharged from hospital with a body weight of 222 kg.

## 3. Discussion

In this case report, we present a styletubation technique for tracheal intubation in a super-super-obese patient (BMI 103 kg/m^2^) undergoing laparoscopic sleeve gastrectomy. Instead of using the conventional DL or VL, we used a styletubation technique for tracheal intubation (Figure 3). The procedure of using a video-assisted intubating stylet was smooth and swift, with high first-pass success (Figure 4). No significant cardiopulmonary instability (e.g., arterial desaturation, hypertension, tachycardia or bradycardia) was observed during intubation. No soft tissue or dental injuries occurred. Although the airway team members did feel a heavier mental load and more stress than during a usual airway exercise, the objective of “safe–accurate–swift” tracheal intubation was satisfactorily achieved in such super-super obesity patients (Table 1).

In general, airway operators are used to adopt either the conventional DL (with the Macintosh blade) or the VL for tracheal intubation in obese adult patients (Table 1). Intuitively, one would think the VL to be superior to the DL in such scenario. The advantages of VL have been demonstrated by proponents of the method and include better glottic visualization, shorter intubating time, less intubation attempts, or higher intubation success rates [20,21,22,23,24,25,26,27,28,29,30]. Therefore, they proposed that the VL be used as the overall first-line tracheal intubation modality versus Macintosh DL in obese patients (and perhaps for overall patient populations to be intubated). In contrast, opposite results regarding such a role for VL appeared in sporadic reports that show either VL is slower [31,32], or that both modalities allow for equally quick and safe airway management [33,34]. 

The overall evidence to support the routine use of VL, particularly in obese patients, is sparse. No definitive study has demonstrated a clear-cut superiority for its routine use in such scenarios. Some previous clinical comparative studies showed inconsistent results in terms of various outcomes such as overall success rates, time to successful tracheal intubation, and number of attempts (e.g., [31,32,33,34]). High-risk obese patients, in comparison to normal subjects, tolerated hypoxia much less and were at a higher risk of aspiration during tracheal intubation. Therefore, both the first-attempt success rate and intubation time become the most important and meaningful key performance indices among all comparators. Understandably, any intubating tools with a better glottis visualization would be advantageous for quicker and successful tracheal intubation. The same true is for such an application in obese populations.

**Table 1 healthcare-11-02256-t001:** Comparison of laryngoscopy and styletubation on tracheal intubation in morbidly obese patients and this patient.

	Laryngoscopy (DL versus VL)	Styletubation (VS) (This Article)
BMI (kg/m^2^)	>40.0 (Moon [11]) 45.9 (Juvin [14]) 43.5 & 42.8 (Marrel [20]) 43 & 44 (Ndoko [21]) 40–43 (Dhonneur [23]) 42.7 & 43.5 (Ranieri [25]) 42 (Yumul [27]) 40.3 (Arslan [29]) 42 (Andersen [31]) 42.5 & 41.2 (Abdallah [32]) 46 (Castillo-Monzon [34]) 43.4 (Gaszynski [42]) 43.7 (Dixit [43]) 48.4 (Riad [44]) 32.9 (Siriussawakul [45]) 34.2 (Siriussawakul [46]) 38.0 (Lavi [47])	103–90.9
Neck circumference (cm)	47.3 & 46.2 (Marrel [20]) 45.5 (Ranieri [25]) 45 (Yumul [27]) 43 (Arslan [29]) 44 (Andersen [31]) 45 (43.4%) (Castillo-Monzon [34]) 42.3 (Riad [44]) 39.0 (Siriussawakul [46])	54
Mallampati class (proportion of class III/IV)	45.0% (Juvin [14]) 32.5% & 32.5% (Marrel [20]) 16% (Ndoko [21]) 21.7–22.6% (Dhonneur [23]) 37.5% & 38.2% (Ranieri [25]) 23.3–40% (Yumul [27]) 7.5% (Arslan [29]) 32% & 22% (Andersen [31]) 27% & 22% (Abdallah [32]) 30.4% & 56.5% (Castillo-Monzon [34]) 42% (Riad [44]) 39.1% (Siriussawakul [45]) 31.4 (Siriussawakul [46]) 30.5% (Lavi [47])	Class IV
Sterno-mental distance (cm)	12.5 (Ranieri [25]) 14.0 (Arslan [29]) 15.0 (Riad [44]) 16.4 (Siriussawakul [45]) 16.4 (Siriussawakul [46])	17
Mouth opening width (interincisor gap) (cm)	<3.5 (26.4%) (Juvin [14]) 4.6 & 4.7 (Marrel [20]) 3.5 (Ndoko [21]) 3.5 (Dhonneur [23]) 3.7 (Ranieri [25]) 5 (Yumul [27]) 4 (Arslan [29]) <4 (4.5% & 9.5%) (Castillo-Monzon [34]) 5.3 (Riad [44]) 5.1 (Siriussawakul [45]) 5.1 (Siriussawakul [46]) <4.0 (18.1%) (Lavi [47])	4.5
Upper lip bite test (proportion of class II/III)	21.7%/5.7% (Siriussawakul [46])	Class II
Pathologically enlarged, swelled, crowding oral cavity, pharynx, or larynx	NA	Crowding surrounding soft tissues; omega-shaped epiglottis
OSAS (Proportion)	35.7% (Juvin [14]) 37.5% & 25% (Marrel [20]) 16% & 30% (Andersen [31]) 43.5% & 34.8% (Castillo-Monzon [34]) 54.3% (Riad [44]) 3.1% (Siriussawakul [46])	Presence
Intubation time: (DA: >10 min)	93 s & 59 s (Marrel [20]) 56 s & 24 s (Ndoko [21]) 69 s & 29 s (Dhonneur [23]) 36.9 s & 13.7 s (Ranieri [25]) 43 s & 22/45/40 s (Yumul [27]) 31 s (Arslan [29]) 32 s & 48 s (Andersen [31]) 26 s & 38 s (Abdallah [32]) 22 s & 17 s (Castillo-Monzon [34]) 1.39 min (Dixit [43]) 45.1 s (Lavi [47])	12 s
Operator’s subjective feeling	83.5% easy (Siriussawakul [45])	Easy, smooth, swift
First-pass success rate	92% & 98% (Andersen [31]) 92% & 86% (Abdallah [32]) 91.3% (Castillo-Monzon [34])	First-pass success
Number of attempts (proportion of more than 1 attempt) (DA: >2 attempts)	20% & 5% (Marrel [20]) 7.5% & 0% (Ndoko [21]) 12.5% & 0% (Ranieri [25]) 2–30% (Yumul [27]) 25% (Arslan [29]) 8% & 2% (Andersen [31]) 8% & 12% (Abdallah [32]) 3.2% & 1.9% for 2 & 3 attempts (Dixit [43])	1 attempt
Cormack–Lehane view (Proportion of III/IV)	10.1% (Juvin [14]) 12% & 0% (Marrel [20]) 20.8% & 0% (Ndoko [21]) 15.1% & 0 (Dhonneur [23]) 7.8% & 0% (Ranieri [25]) 35.4–0% (Yumul [27]) 28% & 4% (Andersen [31]) 22% & 14% (Abdallah [32]) 0% & 4.35% (Castillo-Monzon [34]) 29.3% (Dixit [43]) 7.6%/0.9% (Siriussawakul [46])	NA (POGO 100%)
De-saturation (incidence)	1.9% & 17.0% (Ndoko [21]) 0% (Castillo-Monzon [34]) 1.3% (Siriussawakul [46])	0
Airway injuries (Incidence)	3–36% (Yumul [27]) 0% & 4% (Abdallah [32]) 4% & 22% (Castillo-Monzon [34]) 2.7% (Siriussawakul [46])	0
POST (Incidence)	52.8% & 0% (Ndoko [21]) 81% (Arslan [29]) 32% & 24% (Andersen [31]) 33% & 32% (Abdallah [32]) 4.1% (Siriussawakul [46])	0
IDS score (Proportion)	≥5 (15%) (Juvin [14]) >5 (0% & 20.8%) (Ndoko [21]) >5 (19.7%) (Dixit [43]) >5 (Siriussawakul [45]) >5 (2.3%) (Siriussawakul [46]) >5 (BMI 44.4) (Lavi [47])	NA
DA in MO (Incidence)	4.3% (Moon [11]) 15.5% (Juvin [14]) 15.5% (Arslan [29]) 8% & 12% (Abdallah [32]) 8.7% (Castillo-Monzon [34]) 4.6% (Gaszynski [42]) 2.7% (Dixit [43]) 13.0% (Riad [44]) 14.3% (Siriussawakul [45]) 3.2% (Siriussawakul [46]) BMI 44.4 and IDS > 5 (Lavi [47])	0

DL: direct laryngoscopy; VL: videolaryngoscopy; VS: video-assisted intubating stylet technique; BMI: body mass index; OSAS: obstructive sleep apnea syndrome; DA: difficult airway; POST: post-operative sore throat; IDS: intubation difficulty scale; MO: morbid obesity.

It is worth mentioning that when airway operators use conventional laryngoscopy (both DL and VL) they might still encounter certain difficulties during tracheal intubation. Such technical difficulties and pitfalls include difficulties placing the laryngoscope blade into an oral cavity, injuries to the teeth or soft tissue, an inability to acquire a fair three axis alignment, obstructed visualization of the glottis, and finally difficult/failed advancement of the endotracheal tube into trachea (i.e., you see that you fail) [48,49,50,51]. In contrast, as shown in this case report, styletubation with a video-assisted intubating stylet provides timely success and fulfills the ultimate goal of tracheal intubation in a super-super-obese patient (i.e., swift, accurate, and safe) (Figure 3 and Figure 4; Table 1). In our medical center, we have routinely and universally performed styletubation for tracheal intubation as a daily practice since 2016 [41,52]. It is worth mentioning that such an application of styletubation has also been demonstrated in various potentially difficult airway scenarios, such as for patients with limited cervical mobility, during various ENT procedures, and for COVID-positive patients, etc. [53,54,55,56,57,58]. 

## 4. Conclusions

Many critical issues regarding the peri-anesthesia management of super-super-obese patients undergoing bariatric and non-bariatric surgeries must be considered if we are to resolve the related challenges [59,60]. For example, we must determine the pre-operative optimal positioning of such patients; formulate adequate pre-oxygenation and apneic oxygenation tactics; implement body-weight-adjusted medication regimens; undertake peri-operative monitoring; explain the role of VL and other rescue tools; devise ventilator strategies; promote post-operative care; etc. Currently, there is a move to incorporate the evolving knowledge of optimal airway management in obese patients undergoing major surgery/bariatric surgery into guideline, e.g., a recent consensus statement implemented in Italy [60]. In this case report, we demonstrate that the styletubation technique is applicable for use in a tracheal intubation in a super-super-obese patient undergoing bariatric surgery. In the future, more large-cohort outcome studies in such obese patients undergoing bariatric surgery must be conducted in order to compare various tracheal intubating modalities.

## Figures and Tables

**Figure 1 healthcare-11-02256-f001:**
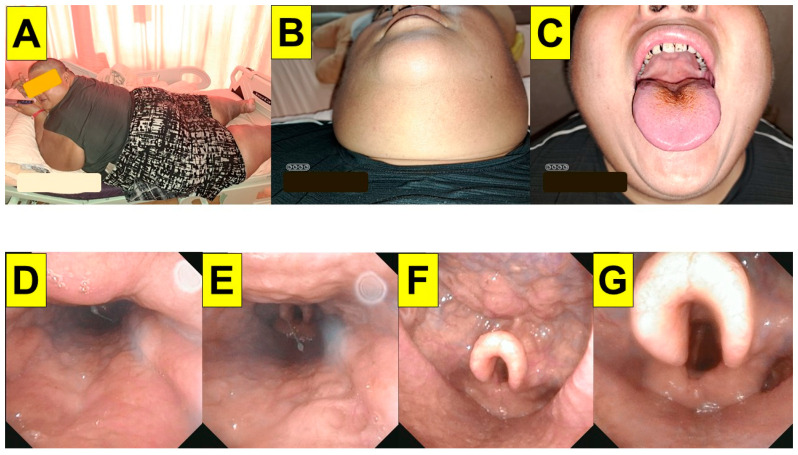
Pre-operative airway evaluation. (**A**): 158 cm, weight: 258 kg, and body mass index (BMI): 103 kg/m^2^). (**B**): neck circumference 54 cm. (**C**): the modified Mallampati classification class IV. (**D**–**G**): serial images from video naso-pharyngo-laryngoscopic examination. An omega-shaped epiglottis is noted.

**Figure 2 healthcare-11-02256-f002:**
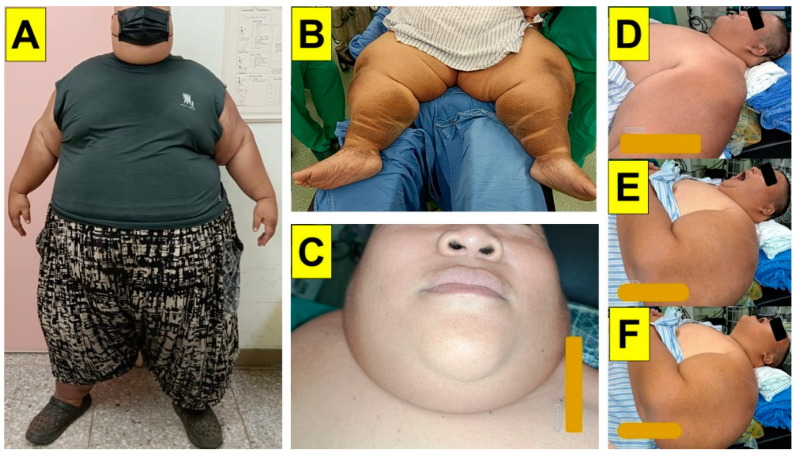
Preoperative drill and planning in the operating room. (**A**): standing posture. (**B**): remained supine position on the operating table. (**C**): the enlarged neck size with thick fat pads. (**D**–**F**): adjusting the height of the pillows and rolls under the patient to line up the ear–sternum in a ramp position. Neutral position with mouth closure (**D**) and opening (**E**). Sniff position with mouth opening (**F**).

**Figure 3 healthcare-11-02256-f003:**
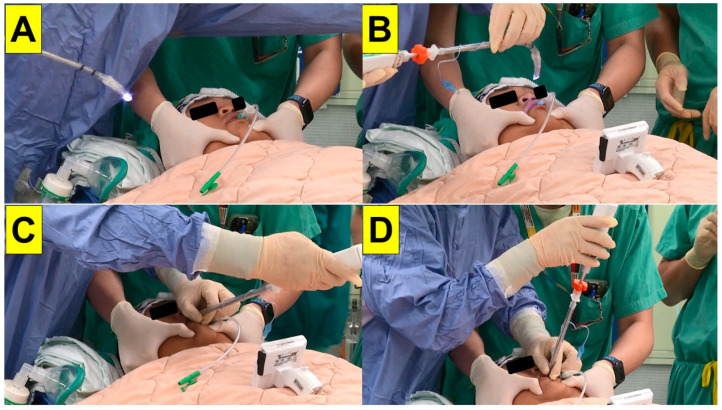
Tracheal intubation conducted via the styletubation technique. (**A**): Before conducting the intubation procedure, the airway assistant helped to open the patient’s mouth. A nasopharyngeal airway-flexible suction tube was applied to clear the airway. (**B**): The airway assistant conducted jaw-thrust maneuver to lift up the patient’s mandible while keeping the airway open. (**C**): The airway operator inserted the video intubating stylet using the guidance shown in a video monitor. (**D**): Glottic visualization before advancing off the endotracheal tube into the patient’s trachea.

**Figure 4 healthcare-11-02256-f004:**
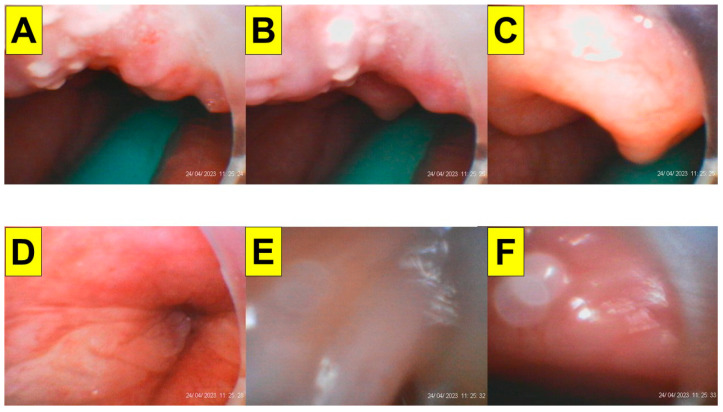
Serial images recorded during styletubation. (**A**): a narrow oro-pharyngeal space. The green naso-pharyngeal airway can be observed. (**B**): the omega-shape epiglottis came to sight. (**C**): a close-up view of the omega-shaped epiglottis. (**D**,**E**): visualization of the glottis and vocal cords. (**F**): view of the tracheal wall before advancement of the endotracheal tube into the trachea. The whole intubation process (from mouth to trachea) was 12 s and succeeded in the first attempt. (See Appendix A).

## Data Availability

Not applicable.

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
