# Peer review of "Styletubation in Bariatric Surgery: A Case Report"

_healthcare, 2023, doi:10.3390/healthcare11162256_

Round 1
Reviewer 1 Report
Thank you for permitting me to review this manuscript
This is a single case study of i successfull ntubation in a superobese patient by stylet
I have few comments after intubation direct laryngoscopy or videolaryngoscopy should have been performed
preoxygenation using a nasal flow oxygen device like optiflow would have been beneficial
this is a case report and the results need to be checked by a series of patient
figure , D and EA are not clear may be images of the supplementary file can be translated in the main article
was muscle relaxation monitored ?
Author Response
MS Title: Styletubation in bariatric surgery
MS Number: 2486050
MS Authors: Wu et al.
Date of Response: 20230718
Reviewer 1
Thank you for permitting me to review this manuscript This is a single case study of i successfull ntubation in a superobese patient by stylet. I have few comments
Q-1: After intubation direct laryngoscopy or videolaryngoscopy should have been performed preoxygenation using a nasal flow oxygen device like optiflow would have been beneficial.
Response-1: Thanks for your comment. Indeed, the High-Flow Nasal Cannula (HFNC) oxygenation does help this kind of super-super obese patient for adequate oxygenation. We therefore applied HFNC for 15 min of pre-oxygenation (described in Page 3, line 101) before induction of anesthesia. Since the practice of tracheal intubation with the styletubation technique is mature during the past decade in our medical center, therefore other supplementary technique (e.g., apneic oxygenation) was not applied. The intubation process only took 12 s and smooth (shown in Figures 3 & 4).
Q-2: This is a case report and the results need to be checked by a series of patient
Response-2: Absolutely. We could not agree with you more. We are aware of the role and limit of a case report. Actually, we are preparing a case series report to reflect our notion and clinical practice experiences with styletubation in various degrees of obese patients who needed tracheal intubation. Also, a prospective randomized clinical study is undergoing conducted by our colleagues. [The Effect of The Use of a Videolaryngoscope and/or Stylet on Intubation Time in Obese Patients], which was registered in https://classic.clinicaltrials.gov/ct2/show/NCT05026671.
Q-3: Figure 4, D and E are not clear may be images of the supplementary file can be translated in the main article.
Response-3: Yes, indeed, the Figure 4-D and 4-E did not look so clear. This is due to the camera lens of the video stylet was very close up to the target at that moment (i.e., the vocal cords). Fortunately, we provided the video-clip as supplementary materials for the readers who can go through this 12-s video clip (the supplementary material has been successfully uploaded).
Q-4: was muscle relaxation monitored ?
Response-4: Yes, by our SOP, we applied Train-of-Four to monitor the intraoperative status of neuromuscular blockade (Page 4, line 105). The ToF monitoring is definitely crucial for such bariatric surgery and provides critical information about the dosage and timing of administering rocuronium and its antagonist (sugammadex). Thank you very much for your constructive comments.

Reviewer 2 Report
In this article, the authors present the usefulness of video-assisted incubating stylet in operating severe obesity patients. It is an interesting report since it focuses on over 103 BMI. However, there are some concerns about this article. 1. The authors should present the operation procedure(sleeve gastrectomy). 2. They should present the patient's state after an operation. 3. To prove the evidence, the authors should compare their facility's conventional and video-assisted methods on severely obese patients.
Moderate English editing is required.
Author Response
MS Title: Styletubation in bariatric surgery
MS Number: 2486050
MS Authors: Wu et al.
Date of Response: 20230718
Reviewer 2
In this article, the authors present the usefulness of video-assisted incubating stylet in operating severe obesity patients. It is an interesting report since it focuses on over 103 BMI. However, there are some concerns about this article.
Q-5. The authors should present the operation procedure (sleeve gastrectomy).
Response-5: Thanks for your excellent comment. We therefore briefly added the description about laparoscopic sleeve gastrectomy in the text (Page 5, line 141).
“In brief, the patient was placed in a supine position and the 5-trocar method was used. A vertical gastrectomy was performed by resection the greater curvature from the distal antrum (5 cm proximal to the pylorus) to the angle of His, using a 36-French bougie as a calibration tube. The resected part of the stomach was then extracted from the periumbilical trocar site. The staple line was reinforced using a running absorbable seromuscular suture.”
Q-6: They should present the patient's state after an operation.
Response-6: Thanks for your comment. The description of post-operative course has already been presented in Page 5, lines 151-158. In addition, the following has been added in the text.
“After 24 hours in the ICU, the patient was discharged to the ward and started receiving the rehabilitation program. On the 7th postoperative day, he was discharged from hospital with a body weight of 222 kg.“
Q-7: To prove the evidence, the authors should compare their facility's conventional and video-assisted methods on severely obese patients.
Response-7: Excellent point!!
Since this submitted manuscript is for a case report, we are aware of the ultimate clinical comparison among various intubating modalities in obese patient populations is of value. This is exactly what we are doing and try to tackle this issue. Our colleagues currently are conducting the following clinical study in our medical center since 2021. The title of this prospective randomized study is: [The Effect of The Use of a Videolaryngoscope and/or Stylet on Intubation Time in Obese Patients], which had been registered in https://classic.clinicaltrials.gov/ct2/show/NCT05026671. Unfortunately, due to the ethics issue, it appears to be difficult to recruit enough voluntary subjects (especially severe obese patients) to enter such clinical study. For this moment, this project was slowly proceeded and extended.

Reviewer 3 Report
This is a case report which focuses on the airway management of a superobese patient undergoing bariatric surgery. There are some issues with this particular manuscript in it's current form, which are summarized in the following point by point comments-
1) The title of the manuscript does reflect it's content. This is a case report of video laryngoscopy being combined with the intubation stylet technique in order to facilitate tracheal intubation in a superobese patient.
2) It is not apparent that the results of a literature review which has been undertaken will also be summarized in the manuscript until this appears in the Discussion section of the manuscript. Of note if the results of a literature review are going to be tabulated it is best if this particular Table appears as a stand alone Table and not in the middle of a discussion section (which is a formatting issue).
3) The way the literature review is summarized into the Table is also problematic. It is difficult to understand what is meant by the varying data elements (some of which appear to have percentages calculated) in each section of the Table. Then there are a combination of black and red dots appearing on each line. The authors need to make it a lot clearer as to what are the actual data elements of interest in each subsection and what the numerical data represent in each case.
4) The literature review is incomplete which constrains the Introduction section along with the Discussion section and the Conclusions. There is now a move to incorporate the evolving knowledge on optimal airway management in the obese patient undergoing major surgery/bariatric surgery into Guidelines. There are now published guidelines out of Italy (the information pertaining to airway management can be found within this particular document) - https://link.springer.com/article/10.1007/s00464-022-09498-y
4) You need to further Discuss the management of this particular case in light of everything which has been published recently (as well as the place of formulating guidelines etc within regions/countries as well as the importance of publishing outcomes on larger cohorts of patients
Author Response
MS Title: Styletubation in bariatric surgery
MS Number: 2486050
MS Authors: Wu et al.
Date of Response: 20230718
Reviewer 3
This is a case report which focuses on the airway management of a superobese patient undergoing bariatric surgery. There are some issues with this particular manuscript in it's current form, which are summarized in the following point by point comments-
Q-8. The title of the manuscript does reflect it's content. This is a case report of video laryngoscopy being combined with the intubation stylet technique in order to facilitate tracheal intubation in a superobese patient.
Response-8: Thanks for your comment. We apologize the confusion regarding the “styletubation” technique itself. As we explained in the Page 1, Lines 15 and Page 2, 61, the styletubation technique is performed simply with a video-assisted intubating stylet by an airway operator. Therefore, it is not designed to combine with laryngoscopes (although it can be under certain conditions). In this particular super-super obese case, we simply applied such a video-assisted intubating stylet without any VL or DL. And this is exactly the interesting point and experience we would like to share.
Q-9. It is not apparent that the results of a literature review which has been undertaken will also be summarized in the manuscript until this appears in the Discussion section of the manuscript. Of note if the results of a literature review are going to be tabulated it is best if this particular Table appears as a stand alone Table and not in the middle of a discussion section (which is a formatting issue).
Response-9: Thanks for your excellent comment on the value and the format of the Table 1. Indeed, it was a tough decision to include the Table 1 in the text of such case report. Our original thought to add the Table 1 in the Discussion section was to provide a quick literature overlook for the readers. Also, there are a plethora of literatures regarding the airway management for obese patients (shown in Table 1). Few among them revealed inconsistent findings, regarding the superiority between DL and VL (e.g., references 31-34). Most important, there are no information about the styletubation, especially for the super-super obese patient. Therefore, with the aid of the Table 1, we hope the readers can easily find the discrepancy among DL/VL/VS for tracheal intubation in obese, super-obese and super-super obese patients.
Q-10. The way the literature review is summarized into the Table is also problematic. It is difficult to understand what is meant by the varying data elements (some of which appear to have percentages calculated) in each section of the Table. Then there are a combination of black and red dots appearing on each line. The authors need to make it a lot clearer as to what are the actual data elements of interest in each subsection and what the numerical data represent in each case.
Response-10: Again we really appreciate your detailed and valuable comments on the Table 1. We first apologize the mess of the format of the Table 1 we created. Now, Table 1 was re-formatted under the new template. Few notes were also added, e.g., proportion/incidence were added when percentages were expressed. Also the colors and dots were corrected or removed as they should be. Hopefully, the new Table 1 now is readable without confusion.
Q-11. The literature review is incomplete which constrains the Introduction section along with the Discussion section and the Conclusions. There is now a move to incorporate the evolving knowledge on optimal airway management in the obese patient undergoing major surgery/bariatric surgery into Guidelines. There are now published guidelines out of Italy (the information pertaining to airway management can be found within this particular document) - https://link.springer.com/article/10.1007/s00464-022-09498-y
Response-11: Thanks for your providing the updated and relevant guideline/consensus paper from Italy. We are aware of our European colleagues made a great effort regarding the issues of bariatric surgeries, including ERAS. Therefore, we added the following two references [59,60] in the revised manuscript (Both views from Belgium and Italy).
- Mulier JP, Dillemans B. Anaesthetic Factors Affecting Outcome After Bariatric Surgery, a Retrospective Levelled Regression Analysis. Obes Surg. 2019 Jun;29(6):1841-1850. doi: 10.1007/s11695-019-03763-1.
- Marinari G, Foletto M, Nagliati C, Navarra G, Borrelli V, Bruni V, Fantola G, Moroni R, Tritapepe L, Monzani R, Sanna D, Carron M, Cataldo R. Enhanced recovery after bariatric surgery: an Italian consensus statement. Surg Endosc. 2022 Oct;36(10):7171-7186. doi: 10.1007/s00464-022-09498-y.
Q-12. You need to further Discuss the management of this particular case in light of everything which has been published recently (as well as the place of formulating guidelines etc within regions/countries as well as the importance of publishing outcomes on larger cohorts of patients
Response-12: Again, we appreciate your comment and suggestion. As the response to Q-11, we have added two important publications to the revised text. Hopefully, the readers can find them useful. The following statement is also added in the Page 9, line 220. Many thanks. The
“In the future, more large-cohort outcome studies in such obese patients undergoing bari-atric surgery need to be conducted to compare various tracheal intubating modalities.”

Reviewer 4 Report
Thank you for the opportunity to review a very interesting article. Case study paper describing a case of a very obese patient who was intubated with a videolaryngoscope. The main objective of the paper was to present the advantages and level of safety of a patient with a large scale of difficulty of intubation. A reliable and richly illustrated description of the use of a stylet tube in a very obese patient undergoing laparoscopic sleeve gastrectomy is a useful and easy-to-learn technique. There are some issues that need to be considered.
-
In the description, the names of drugs should be capitalized,
-
Please indicate the data collection protocol and procedures for anthropometric measurements and who performed them?
-
In the description of Figure 4, the authors should add information about which supplementary materials,
-
Identify the authors of clinical comparative studies that showed inconsistent results in overall success rate, time to successful endotracheal intubation, number of endotracheal intubation attempts and number of intubation attempts.
Author Response
MS Title: Styletubation in bariatric surgery
MS Number: 2486050
MS Authors: Wu et al.
Date of Response: 20230718
Reviewer-4
Thank you for the opportunity to review a very interesting article. Case study paper describing a case of a very obese patient who was intubated with a videolaryngoscope. The main objective of the paper was to present the advantages and level of safety of a patient with a large scale of difficulty of intubation. A reliable and richly illustrated description of the use of a stylet tube in a very obese patient undergoing laparoscopic sleeve gastrectomy is a useful and easy-to-learn technique. There are some issues that need to be considered.
Q-13. In the description, the names of drugs should be capitalized,
Response-13: Thanks for your reminding regarding the names of medications be capitalized. We were instructed that all the names of medications in the text should be capitalized only when they are of tradenames. We will contact the editorial office to confirm this issue. Many thanks.
Q-14. Please indicate the data collection protocol and procedures for anthropometric measurements and who performed them?
Response-14: Thanks for the correction. The author (H.-N Luk) is the team member of the bariatric surgery group in Tzuchi Medical Center and also served as the leading anesthesiologist for the special task of bariatric surgery (the leader is Dr. B.-G. Wu). Therefore, the pre-anesthesia evaluation task (including the standard airway evaluation protocol) was completed by Dr. Luk himself.
Q-15. In the description of Figure 4, the authors should add information about which supplementary materials.
Response-15: Apologize for the mistake. The statement has been added in the legend of the Figure 4 “(See Video S1 in the Supplementary Materials)”. The video clip has been successfully uploaded to the Journal website.
Q-16: Identify the authors of clinical comparative studies that showed inconsistent results in overall success rate, time to successful endotracheal intubation, number of endotracheal intubation attempts and number of intubation attempts.
Response-16: Thanks for your great points. As we stated in Page 6, lines 171-172, some clinical reports did not consistently show the advantages of VL over DL (e.g., [31, 32] or [33, 34]).
The following “inconsistent” results reports found in literature {31-34} when DL was compared to VL (Table 1):
- The intubation time: DL is quicker than VL (Andersen, [31]; Abdallah [32])
- First-pass success rate: DL is higher than VL (Abdallah [32])
- Overall number of attempts: DL has less attempt to success than VL (Abdallah [32])

Round 2
Reviewer 1 Report
The authors have provided fair answers to my concerns , but I prefer to wait dfor their case series which will be more beneficial
Author Response
MS Title: Styletubation in bariatric surgery
MS Number: 2486050
MS Authors: Wu et al.
Date of Response: 20230727
Reviewer 1
Comment 1: The authors have provided fair answers to my concerns , but I prefer to wait dfor their case series which will be more beneficial
Response 1: We appreciate your encourage on this issue. Indeed, we continuously pushed ourselves to finish our case series report as soon as possible. Hopefully, our report can come up with a larger picture on this issue.

Reviewer 3 Report
There are still some issues with the revised version of the manuscript-
1) The title of the manuscript needs to reflect the fact that you are reporting on one case-ie it is a case report
2) Table 1 contains minimal information for what is a review of the literature, ie there is no metric for the denominator for the numbers of patients in each of the studies quoted. Hence the reader is unable to ascertain how much published evidence there really is (unless you go and read each individual reference yourself)!
3) Despite adding some references there is no discussion on the move to establish guidelines in some regions (despite the comments made to the reviewer)
Author Response
MS Title: Styletubation in bariatric surgery
MS Number: 2486050
MS Authors: Wu et al.
Date of Response: 20230727
Reviewer 3
There are still some issues with the revised version of the manuscript-
Comment 1: The title of the manuscript needs to reflect the fact that you are reporting on one case-ie it is a case report
Response 1: Thank you for your suggestion. Therefore, the Title has been revised as follows:
“Styletubation in Bariatric Surgery: A case report”
Comment 2: Table 1 contains minimal information for what is a review of the literature, ie there is no metric for the denominator for the numbers of patients in each of the studies quoted. Hence the reader is unable to ascertain how much published evidence there really is (unless you go and read each individual reference yourself)!
Response 2: Thank you very much for your excellent common on this issue. Indeed, there are no metrics for denominator of each presented data in the Table 1. This is simply because of the format and the scale of the Table itself. Therefore, we consistently used “percentage” to give the readers some quantitative ideas on particular issues. We expect that the readers will explore the actual figures in the referred literature if they found such numbers are interesting and worthy to go deeper. Again, this is simply a case report and not possible to provide full-range data as the meta-analysis could do. We apologize for this weakness of the presentation.
Comment 3: Despite adding some references there is no discussion on the move to establish guidelines in some regions (despite the comments made to the reviewer)
Response 3: Thank you for your comment on the guideline issue. We also believe the need to have such guideline or consensus paper is helpful and important locally and globally. The following statement was added into the revised text therefore.
“Currently, there is a move to incorporate the evolving knowledge on optimal airway management in the obese patient undergoing major surgery/bariatric surgery into guideline, e.g., a recent consensus statement implemented in Italy [60].”
